# Morphological traits and machine learning for genetic lineage prediction of two reef-building corals

Guinther Mitushasi[1]*, Yuko F. Kitano[2], Nicolas Oury[3¤a], Hélène Magalon[3],
David A. Paz-García[4], Eric Armstrong[5], Benjamin C. C. Hume[6], Barbara Porro[7],
Clémentine Moulin[8], Emilie Boissin[9], Guillaume Bourdin[10], Guillaume Iwankow[11],
Julie Poulain[12], Sarah Romac[11], Maggie M. Reddy[13], Tara Pacific Consortium Coordinators[¶],
Serge Planes[9], Denis Allemand[14], Christian R. Voolstra[6], Didier Forcioli[15], Sylvain Agostini[1¤b*]

1 Shimoda Marine Research Center, University of Tsukuba, 5-10-1 Shimoda, Shizuoka, Japan, 2 Japan Wildlife Research Center, Tokyo, Japan, 3 UMR ENTROPIE (UMR – Université de La Réunion, IRD, IFREMER, Université de Nouvelle-Calédonie, CNRS), Université de La Réunion, St Denis, La Réunion, France, 4 Centro de Investigaciones Biológicas del Noroeste (CIBNOR), Laboratorio de Genética para la Conservación, Av. IPN 195, Col. Playa Palo de Santa Rita Sur, La Paz, Baja California Sur, México, 5 PSL Research University, EPHE, CNRS, Université de Perpignan, Perpignan, France, 6 Department of Biology, University of Konstanz, Konstanz, Germany, 7 French National Institute for Agriculture, Food, and Environment (INRAE), Université Côte d'Azur, ISA, France, 8 Fondation Tara Océan, Base Tara, 8 rue de Prague, 75 012, Paris, France, 9 PSL Research University: EPHE-UPVD-CNRS, USR CRIOBE, Laboratoire d'Excellence CORAIL, Université de Perpignan, Perpignan, France, 10 School of Marine Sciences, University of Maine, Orono, Maine, United States of America, 11 Sorbonne Université, CNRS, Station Biologique de Roscoff, AD2M, UMR, ECOMAP, Roscoff, France, 12 Research Federation for the study of Global Ocean Systems Ecology and Evolution, FR2022/Tara GOSEE, 3 rue Michel-Ange, Paris, France, 13 School of Biological and Chemical Sciences, Ryan Institute, University of Galway, University Road, H91, Galway, Ireland, 14 Centre Scientifique de Monaco, 8 Quai Antoine Ier, MC-98000, Monaco, Principality of Monaco, 15 LIA ROPSE, Laboratoire International Associé Université Côte d'Azur—Centre Scientifique de Monaco, Monaco, Principality of Monaco,

¶ Membership of the Tara Pacific Consortium is provided in the Acknowledgments.
¤a Marine Science Program, Biological and Environmental Science and Engineering (BESE) Division, King Abdullah University of Science and Technology (KAUST), Thuwal, Saudi Arabia
¤b UMR Entropie, IRD, Université de la Réunion, IFREMER, CNRS, Université de Nouvelle-Calédonie, Nouméa, New Caledonia.
* allblueguinther@gmail.com (GM); sylvain.agostini@ird.fr (SA)

## Abstract

Integrating multiple lines of evidence that support molecular taxonomy analysis has proven to be a robust method for species delimitation in scleractinian corals. However, morphology often conflicts with genetic approaches due to high phenotypic plasticity and convergence. Understanding morphological variation among species is crucial to studying coral distribution, life history, ecology, and evolution. Here, we present an application of Random Forest models for coral species identification based on morphological annotation of the corallum and corallites. We show that the integration of molecular and morphological trait analysis can be improved using machine learning. Morphological traits were documented for *Porites* and *Pocillopora* coral species that were collected and genotyped through genome-wide, genetical hierarchical clustering, and coalescence analyses for the Tara Pacific Expedition.

**Data availability statement:** The resources used here including images, morphological annotations, code, and statistical analysis are available in a Zenodo repository at 10.5281/zenodo.14666394 and 10.5281/zenodo.14650668. The in-situ images of the coral colonies taken during the Tara Pacific expedition are available in the PANGAEA repository at https://store.pangaea.de/Projects/TARA-PACIFIC/Images/

**Funding:** Work from GM is supported by JST SPRING (https://www.jst.go.jp/jisedai/spring/en/outline/index.html), Grant Number JPMJSP2124. The funders had no role in study design, data collection and analysis, decision to publish, or preparation of the manuscript. There was no additional external funding received for this study.

**Competing interests:** The authors have declared that no competing interests exist.

While *Porites* only included three tentative species, most *Pocillopora* species were accounted by included specimens from the western Indian Ocean, tropical Southwestern Pacific, and southeast Polynesia. Two Random Forest models per genus were trained on the morphological annotations using the genetic lineage labels. One model was developed for in-situ image identification and used corallum traits measured from in-situ photographs. Another model for integrative species identification combined corallum and corallite data measured on scanning electron micrographs. Random Forest models outperformed traditional dimension reduction methods like PCA and FAMD followed by k-means and hierarchical clustering by classifying the correct genetic lineage despite morphological clusters overlapping. This machine learning approach is reproducible, cost-effective, and accessible, reducing the need for taxonomic expertise. It can complement molecular and phylogenetic studies and support image identification, highlighting its potential to advance a coral integrative taxonomy workflow.

## Introduction

Molecular taxonomy studies have contributed to a better understanding of scleractinian coral systematics, while challenging conventional morphology-based taxonomy [1,2]. Integrative taxonomy combines multiple lines of evidence—such as genetics, reproduction, symbiont association, ecology, and morphology [3–6] — and is considered a robust approach for species delimitation [7,8]. Among the different lines of evidence, morphology is essential for understanding the physical and behavioral interactions, life history, adaptation, and evolution of different species. However, the phenotypic plasticity observed in many hermatypic coral genera makes the alignment of morphological traits with other lines of evidence challenging [9]. To provide more resolutive systematic approaches, the investigation of morphological traits using alternative and reproducible methods should be considered.

Congruencies between morphology and genetic data are often investigated using multivariate dimension reduction analyses, such as principal component analysis (PCA) or factor analysis of mixed data (FAMD), followed by clustering methods — k-means, hierarchical clustering, and discriminant analysis of principal components. Machine-learning algorithms have been used for classification, prediction, and modeling operations [10,11]. Among the many algorithms available, the Random Forest (RF) [12] classifies based on the majority of votes generated from multiple decision trees with a random bootstrap of sampling and predictors [13]. RF has been applied for classification across different fields, including remote sensing [14], genetics [15], marine ecology [16,17], and others (see Fawagreh et al. [18] for review). RF has also been found to surpass other linear, decision-tree, and classification methods when strong interactions among variables are found [11,13]. It can be used for highly multivariate dimensional data that combine both quantitative and qualitative variables, is less prone to

overfitting owing to its bagging and randomness features, and does not require data normalization [12,18]. To the best of our knowledge, RF and morphological annotation have not yet been used to classify hermatypic corals. The features and usefulness of such algorithms could be tested as alternative methods for the morphological analysis of scleractinian corals.

Hermatypic coral morphology is traditionally assessed through phenotypic variations observed in the macro or colony (corallum) and micro (corallite) skeletal morphologies [19]. A reliable approach for the analysis of these morphological traits would allow for the identification of corals in the field, avoid sampling errors and inaccurate biodiversity assessments, and further reveal their interactions with the environment [20]. At the corallum level, corals exhibit multiple morphotypes across and within species. Several studies have quantified corallum morphology using in-situ and ex-situ 3D modeling techniques [21,22], geometric morphometrics [23], and linear 2D measurements of colonies or fragments [24,25]. These methods often require the sampling of large fragments and entire colonies, long diving periods, and large amounts of processing power, making these unsuitable for noninvasive and large ecological surveys. Promising results have been obtained for the identification of coral species without damaging ecosystems using recent deep-learning image analysis software [26,27]. Computer vision and deep-learning image analysis rely on the color and texture of images and do not provide information on morphological traits, thereby limiting the investigation of phylogeny and coral evolution. Corallite morphology often aligns with genetics in many genera compared to corallum morphology [28]. Linear and geometric morphometrics have linked genomics with micromorphology across families, genera, and species [4,23,29,30]. The widely studied *Pocillopora* genus has shown incongruence between corallum morphology and genetic lineage identification [24,31,32], while micromorphological characters and genotypes down to the species level seems to be more congruent [6,33,34]. Micromorphology analysis often relies on the qualitative assessment of traits, and the complexity or lack of measurable parameters makes their description open to interpretation and difficult to reproduce. Thus, combining a large number of quantitative parameters and alternative classification methods could allow the detection of the underlying variation in skeletal traits at the species level.

The corals used here were collected at 11 islands across the Pacific Ocean during the Tara Pacific Expedition described by Voolstra et al. [5], and complemented by *Pocillopora* corals included and described by Oury et al. [6], sampled from the western Indian Ocean, tropical southwestern Pacific, and southeast Polynesia. All the samples were assigned to species using genome-wide genetic variation data, genetical hierarchical clustering, and coalescence analyses [5,6]. The *Porites* samples were assigned to three genetic lineages: *P. evermanni*, hereafter referred to as SSH1_ pever, and two new cryptic lineages within *Porites lobata* species, not described in Hellberg et al. [35], referred to as SSH2_plob and SSH3_plob [5]. *Pocillopora* samples were found to belong to thirteen genetic lineages – sensu Oury et al. [6]. The lineages are: GSH09c (*P. grandis* [34]), GSH01 (*P.* cf. *effusa* [34]), GSH09b (*P. meandrina* [34]), GSH13c (*P. verrucosa* [34]), GSH05 (*P. acuta* [34]), GSH10 (*P.* cf. *brevicornis* [34]), and GSH04 (*P. damicornis* [34]), GSH14 (*P. tuahiniensis* [36]), GSH13b (*P. villosa nomen nudum* [37]). According to Oury et al. [6] GSH09a correspond to a distinct species from *P. meandrina* referred to here as *P.* aff. *meandrina*. GHS13a is a species undescribed or incorrectly synonymized with *P. verrucosa*, here referred to as *P.* aff. *verrucosa*. GSH12 is a hybrid between GSH13c and GSH13a here referred to as *P. verrucosa x P.* aff. *verrucosa*. GSH15 is a hybrid between GSH13c and GSH14 referred to as *P. verrucosa x P. tuahiniensis*.

Linear morphometrics were applied to the corallum and corallite morphologies of two widespread coral genera (*Porites* and *Pocillopora*) to generate (1) a model for in-situ image identification of coral species using corallum morphology and (2) a model for species identification using both corallite and corallum morphological traits. We also discuss the usefulness of the proposed approach in comparison with commonly used analyses to investigate coral species delimitation using morphology. The RF models were used to classify and predict coral species based on their acquired morphological traits.

## Materials and methods

### Sampling

The corals used here were collected during the first year of the Tara Pacific Expedition (July 2016 to February 2017). Sampling includes three sites on 11 islands – Islas de las Perlas, Coiba, Malpelo, Rapa Nui, Ducie, Gambier, Moorea, Aitutaki, Niue, Upolu, and Guam – across an 18,000 km East-West transect. Two target species, *Porites lobata* and *Pocillopora meandrina*, were sampled based on colony morphology [5]. Coral colony in-situ photographs and coral fragments from a minimum of three colonies per site were used for morphological measurements. Genotyping information for the Tara Pacific samples was obtained from Voolstra et al. [5]. To include the majority of known *Pocillopora* species, another 214 *Pocillopora* colonies with in-situ colony images, colony fragment photographs, scanning electron microscopy (SEM) of the corallite structure, and genotyping information were obtained from Oury et al. [6].

### Ethics statement

The coral colonies of *Pocillopora* spp. and *Porites* spp. collected during the Tara Pacific Expedition between July 2016 and February 2017 were sampled in accordance with UNCLOS and CITES permits (see S1 Note). Detailed information concerning the sampling localities, handling, metadata, and sampling protocols can be found in Lombard et al. [38]. Additional samples used were obtained from the publication of Oury et al. [6].

### Morphological measurements

After sampling, the coral fragments were placed in a 3–4% bleach solution on board the Tara schooner for a minimum of two days, rinsed with tap water, and kept dry in falcon tubes. Additional washing steps using a 10% bleach solution for 12 h were conducted prior to morphological analysis for further tissue removal. The samples were rinsed with freshwater, oven-dried for a minimum of 8 h, and then placed in a vacuum chamber for a minimum of 3 h to remove excess moisture or air trapped in the interior of the fragments before SEM imaging for micro (corallite) morphological annotations. Scanning electron microscopy was conducted using a JCM-5000 NeoScope™ Tabletop SEM (Japan) without the need of coating. Macro (corallum) morphology measurements were conducted using in-situ colony photographs, as described by Voolstra et al. [5]. Verruca measurements of *Pocillopora* colony fragments were included in the corallum morphological analysis. These were conducted on fragment photographs taken with an Olympus TG5 camera with a 0.2 mm-diameter fishing line for scaling.

Porites morphometric measurements were done following Forsman et al. [23]. Linear measurements of *Pocillopora* spp. were conducted based on previous studies [6,25,34]. Additional parameters were included for both genera. The names, landmarks and full descriptions of all morphological parameters are described in Table 1 and 2, and an illustration of the measurements are found in Fig 1 and 2. Manual annotations of coral morphology on the *Porites* and *Pocillopora* corallum and corallite structures were performed using ImageJ v1.50e [39], and the image annotation plugin objectj v1.05j [40]. Corallum measurements, were conducted on one whole colony image per sample [5], and one branch fragment image per sample in the case for *Pocillopora* fragments. The corallite measurements were performed using a minimal number of three SEM images, and three corallites per sample.

### Random forest models

RF classification and associated statistical analyses were conducted using R programming language [41] and the following packages: *randomForest* [42], *vegan* [43], and *tidyverse* [44]. RF is a decision-tree machine-learning-based method for classification and regression analysis [12]. Unsupervised RF has been used to identify morphotypes in accordance with genetic lineages [5]. Here, the RF model was trained with genetic lineage labels for supervised classification. The algorithm builds multiple decision trees (ntree) generated from two bootstrap subsets of observations that are randomly

**Table 1. Morphological measurements of *Porites* spp.** See Fig 1 for an illustration of parameters.

| Parameters | Landmarks | Description |
| --- | --- | --- |
| **Corallum morphology** | | |
| COLPS | (B) 2 | Projected surface of the colony (mm$^2$). |
| COLDIAM | (B) white segmented lines | Average of four diameters crossing the centroid of the projected surface of the colony (mm). |
| LD | (B) - | Density of lobes computed on the number of lobes in a 20 × 20 cm quadrat drawn on the colony photo using the ImageJ rectangle tool, divided by the quadrat area (count/mm$^2$). |
| PR* | (A) 1 | Presence or absence of ridges on the surface of the corallum. |
| GF* | (A) - | Type of growth form (columnar, massive, or encrusting). |
| **Corallite morphology** | | |
| CAS | (C) 61–62, 61–63, 61–64, 61–65 | Calice spacing between the center of a randomly selected columella and the center of up to eight columellae from adjacent calices (mm). |
| CAW | (C) 1–13, 3–15, 5–17, 7–19, 9–21, 11–23 | Calice width calculated as the distance means between six pairs of opposite landmarks representing the base of each septum (mm). |
| CAA | (C) 1-3-5-7-9-11-13-15-17-19-21-23-1 | Calice area determined by a polygon formed by landmarks at the base of each septum (mm$^2$). |
| THET | (C) 1–49, 3–50, 5–51, 7–52, 9–53, 11–54, 13–55, 15–56, 17–57, 19–58, 21–59, 23–60 | Theca thickness calculated as the distance averages between 12 landmark pairs. A pair is composed of a point at the base of one septum, and its closest point at the inner part of the theca of an adjacent calice (mm). |
| SEPL | (C) 1–2, 3–4, 5–6, 7–8, 9–10, 11–12, 13–14, 15–16, 17–18, 19–20, 21–22, 23–24 | Septum length calculated as the average of the lengths of all septa (mm). |
| SEPW | (C) 25–26, 27–28, 29–30, 31–32, 33–34, 35–36, 37–38, 39–40, 41–42, 43–44, 45–46, 47-48 | Septum width calculated as the average of the widths of all septa (mm). |
| SEPS | (C) 1–3, 3–5, 5–7, 7–9, 9–11, 11–13, 13–15, 15–17, 17–19, 19–21, 21–23, 23–1 | Septa spacing calculated as the distance averages between landmarks at the base of each adjacent septum (mm). |
| SEP-SVENA | (C) 13–11 | Space between the base of the ventral directive septum and the base of the clockwise antecessor septum (mm). |
| SEP-SVENS | (C) 13–15 | Space between the base of the ventral directive septum and the base of the clockwise successor septum (mm). |
| DORL | (C) 1–2 | Length of the dorsal directive septum (mm). |
| VENL | (C) 13–14 | Length of the ventral directive septum (mm). |
| FA | (C) 2-4-6-8-10-12-14-16-18-20-22-24-2 | Area of fossa determined by a polygon formed by landmarks represented by the tip end of each septum (mm$^2$). |
| FW | (C) 2–14, 4–16, 6–18, 8–20, 10–22, 12–24 | Fossa width calculated as the average of distances between six pairs of opposite landmarks representing the tip end of each septum (mm). |
| PN | (D) a | Number of pali per calice (count/n). |
| DDPW | (D) 1–2 | Dorsal directive pali width (mm). |
| VDPW | (D) 3–4 | Ventral directive pali width (mm). |
| LPW | (D) 5–6 | Lateral pali width calculated as the average of the four lateral pali widths (mm). |
| RN | (D) b. | Number of radi per calice (count/n). |
| COPS | (D) yellow line | Projected surface of the columella (mm$^2$). |

* Qualitative parameters

**Table 2. Morphological measurements of *Pocillopora* spp.** See Fig 2 for an illustration of parameters.

| Parameters | Points | Description |
|---|---|---|
| **Corallum morphology** | | |
| COLPS | (A) 8 | Projected surface of the colony (mm$^2$). |
| COLDIAM | (A) white segmented lines | Average of four diameters crossing the centroid of the projected colony surface (mm). |
| TD | (A) - | Branch tip density calculated from the division of the number of tips on the COLPS and its area (count/mm$^2$). |
| BS | (A) 1–2 | Branch spacing between a randomly selected center branch and minimum of three adjacent branches (mm). |
| TLL | (A) 5–6 | Tip length line: a straight line connecting the two extremities of the branch top (mm). |
| TLP | (A) 7 | Tip length polyline: a line connecting the two extremities of the branch top following the tip curvature (mm). |
| TLR | (A) - | Ratio between the TLP and TLL to detect the meander shape degree of the branch tip. |
| TW | (A) 3–4 | Width of the tip (mm). |
| COLR | (A) - | Colony roundness calculated from the difference between the maximum and minimum diameter divided by two. |
| VA | (B) 5-6-7 | Verruca angle in relation to the branch axis (degrees). |
| VS | (B) 3–4 | Verrucae spacing between a randomly selected center verruca and adjacent verrucae (mm). |
| VL | (B) 5–6 | Verruca length (mm). |
| VW | (B) 1–2. | Verruca width (mm). |
| VD | (B) - | Verruca density calculated from the number of verrucae on the projected surface of one fragment and then divided (count/mm$^2$). |
| **Corallite morphology** | | |
| CAS | (E) 1–2, 1–3, 1–4, 1–5, 1–6 | Calice spacing between the center of a randomly selected columella and up to five columellae from adjacent calices (mm). |
| CADIAM1 | (C) 5–6 | Calice diameter measured from the base of the dorsal directive to the base of the ventral directive septa (mm). |
| CADIAM2 | (C) 7–8 | Calice diameter measured as perpendicular to CADIAM1 |
| SEPL | (C) 1–2 | Length of four lateral septa (mm). |
| SEPW | (C) 3–4 | Width of four lateral septa (mm). |
| SEPN | (C) - | Number of septa per calice (count/n). |
| CODIAM | (C) white segmented lines. | Average of four diameters crossing the centroid of the projected columella surface (mm). |
| CODIAMF | (C) - | Feret's diameter of the projected columella surface (mm). |
| COPS | (C) red segmented line | Projected columella surface (mm$^2$). |
| COR | (C) - | Columella roundness calculated from the difference between the COADIAMF and minimum diameter divided by two. |
| STN | (D) S | Number of stylae per calice (count/n). |
| SPIS | (C) 11–12 | Spinule spacing measured from the top center of spinules on the surface of the corallum. Spinules present on the verruca were not considered (mm). |
| SPIW | (C) 9–10 | Width of spinules (mm). |

selected from the original data, training dataset, and out-of-bag testing dataset. At each node, a different set of variables (*mtry*) was used for the best binary split. Training dataset trees were used to predict the out-of-bag observations. The final class prediction accuracy and model out-of-bag error rate (OOB) were calculated by averaging all observations from each out-of-bag sample. The OOB is an unbiased internal estimate of the model's generalization error, classification strength and dependence [12]. This algorithm was applied for the species classification of the two coral genera—*Porites* and *Pocillopora*—based on the manual morphological annotations described in Table 1 and 2. The genetic lineage labels

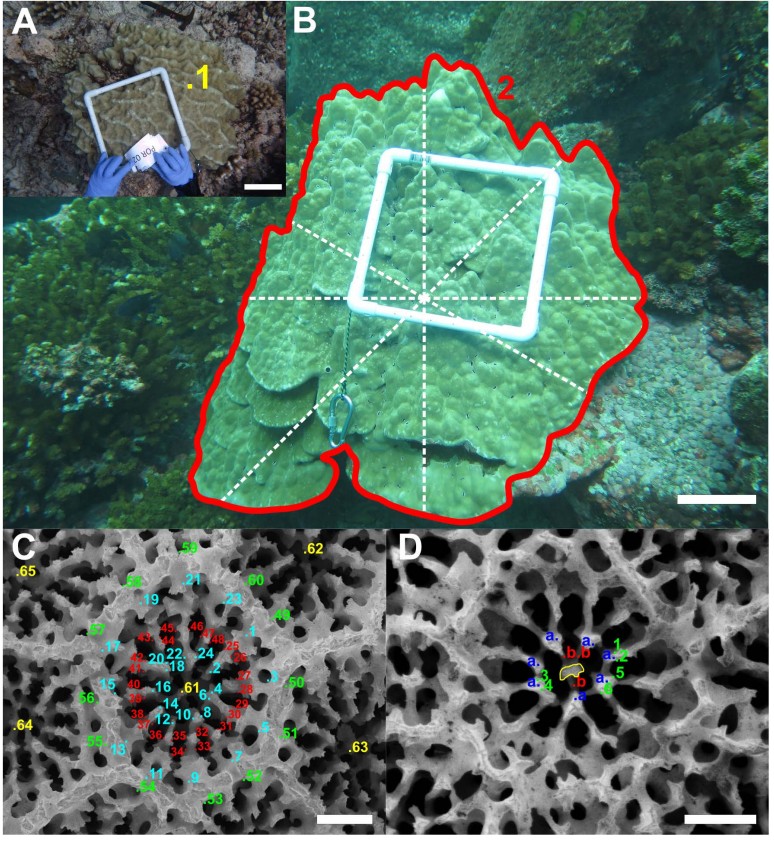

**Fig 1. Example of *Porites* spp. morphological measurements.** In-situ corallum (A-B) and SEM corallite (C-D) measurements. See Table 1 for parameters and landmark descriptions. A and B scale bars are 10 cm. C and D scale bars are 0.5 mm.

for each sample used during training are correspondent to Voolstra et al. [5] for *Porites* (*P. evermanni*, here referred to SSH1_pever, SSH2_plob, and SSH3_plob), and to Oury et al. [6] for *Pocillopora* (GSH09c, GSH14, GSH01, GSH09b, GSH09a, GSH13c, GSH13a, GSH12, GSH15, GSH13b, GSH05, GSH10, and GSH04). Two models were developed for each genus. The first used corallum morphological measurements and the second combined corallum and corallite morphologies. RF tuning (randomForest::tuneRF) was performed to evaluate the optimal number of variables used at each splitting node of each decision tree.

Determining an ideal *mtry* can increase model performance, resulting in a lower OOB error and increased model prediction [12,45]. RF tuning iterated multiple RF models with different numbers of *mtry*. Finally, the model (ntree = 500) with the least number of OOB errors was generated. A confusion matrix of the species classification and OOB error was obtained to assess the model performance. Principal coordinate analysis ordination (PCoA; stats::cmdscale; R Core Team, [41]) generated from the RF proximity matrix was used to visualize the performance of the model and morphological variance in compliance with genetic lineages. The prediction confidence for each sample was investigated by extracting the number of votes given to the correct genetic lineage during the decision-tree building process—standardized across samples using the function vegan::decostand. Samples were also labelled into two groups: correctly classified (CC) and wrongly classified (WC) corals, depending on whether the lineage with the highest vote corresponded to the genetic lineage determined. A pairwise PERMANOVA (metaMisc::adonis_pairwise; Mikryukov, [46]; 999 permutations; P-adjusted Hochberg) was applied on the RF proximity matrix to test for the significance of morphological variances

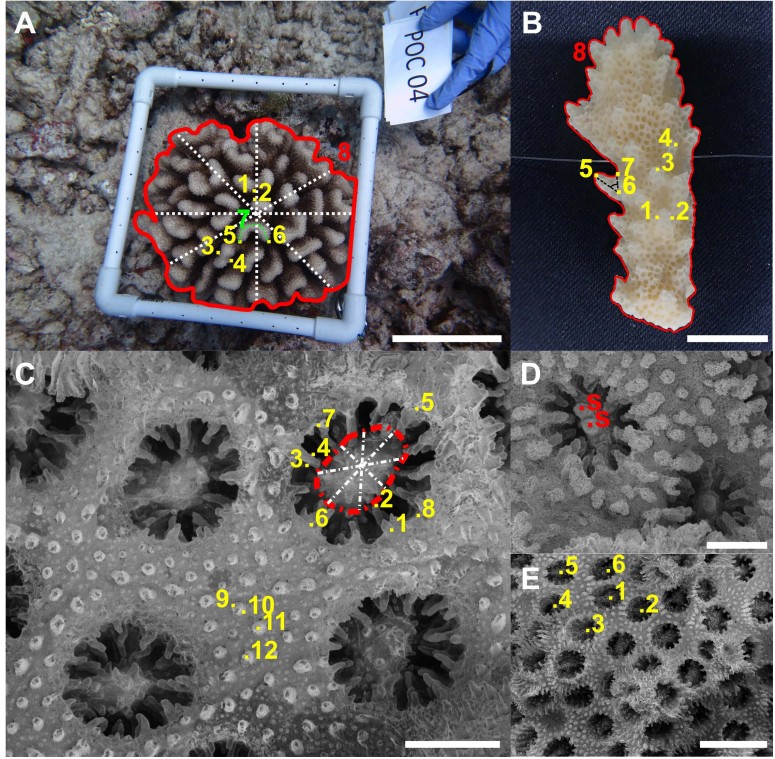

**Fig 2. Example of *Pocillopora* spp. morphological measurements.** In-situ (A) and branch fragment (B) corallum images. SEM (C-E) corallite images. See Table 2 for parameters and landmarks descriptions. Scale bars are 10 cm (A), 10 mm (B), 0.5 mm (C-D), and 2 mm (E).

between species. The mean decrease accuracy (MDA) and mean decrease Gini (MDG) scores extracted from the different models were used to evaluate the predictive power of morphological variables and node homogeneity.

## Dimension reduction and clustering analysis

To further investigate the performance of the RF algorithm compared to traditional statistical analysis, two widely used dimension reduction and clustering analysis were conducted on raw morphological annotation datasets. Each method was selected based on the type of data produced. As *Porites* annotations had both quantitative and qualitative measurements, factor analysis of mixed data (FactoMineR::FAMD; Lê et al. [47]) and hierarchical clustering (stats::hclust; R Core Team [41]) were performed. *Porites* hierarchical clustering was conducted on the morphological annotations distance matrices produced using the Gower method (kmed::distmix; Budiaji, [48]) where the number of clusters were set as the number of genetic lineages present in the data set ($k = 3$). *Pocillopora* annotations were all quantitative, therefore a PCA (stats::prcomp; R Core Team, [41]) and k-means clustering (stats::kmeans; R Core Team, [41]) using the algorithm of Hartigan and Wong [49] were conducted. K-means clustering was performed on the matrices of the raw morphological annotations where the number of clusters were set as the number of genetic lineages present in the data set ($k = 13$). The morphological spaces described by these analyses were visualized by highlighting CC and WC coral samples by the RF models. To test the significance of morphological variances among species obtained by FAMD and PCA, a pairwise PERMANOVA with 999 permutations and Hochberg *P*-adjusted were applied to the distance matrices of the morphological annotations. The *Porites and Pocillopora* distance matrices were calculated using the Gower method for mixed data and Euclidean distances (vegan::vegdist; Oksanen et al. [43]), respectively.

The resources used here including images, morphological annotations, code and statistical analysis are available in a Zenodo repository [50,51]. The in-situ images of the coral colonies taken during the Tara Pacific expedition can be found at https://store.pangaea.de/Projects/TARA-PACIFIC/Images/.

## Results

### *Porites* spp. models

The *Porites* corallum RF model showed an out-of-bag error of 35.08%, resulting in classification accuracies of 73.5% for SSH2_plob, 71.4% for SSH3_plob, and 16.7% for SSH1_pever (Fig 3A). The pairwise PERMANOVA of the *Porites* corallum proximity matrix obtained by the RF model was significant among all comparisons ($p \leq 0.024$, S1 Table). Principal coordinate analysis (PCoA) of the proximity matrix revealed at least two distinct clusters containing corals of all three genetic lineages. Independent of the overlap, the model correctly classified SSH2_plob and SSH3_plob corals (S1A Fig). Of the five parameters included, the three most relevant according to their MDA and MDG scores were the growth form (GF), presence of ridges (PR), and average of four colony diameters (COLDIAM) (S1C Fig). In comparison, FAMD on the same annotation data showed one major cluster containing all three genetic lineages and another consisting of a few SSH3_plob corals, determined by their columnar growth form (S1B-D Fig). The distribution of correctly and wrongly classified specimens according to the RF model seemed to be independent of clustering by the FAMD morphospace (S1B Fig). Similarly to FAMD, the PCoA on the Gower distance grouped by hierarchical clustering showed three clusters that included individuals from all three lineages, and one that was homogeneous to SSH3_plob (S2A Fig). The pairwise PERMANOVA computed for the Gower distance using the same annotation data was significant only between SSH2_plob and SSH3_plob ($F_{1,67} = 6.56$, $p = 0.006$; S1 Table).

*Porites* corallum and corallite models achieved a 12.16% out-of-bag error with classification accuracies of 94.1% for SSH3_plob, 83.9% for SSH2_plob, and 77.8% for SSH1_pever (Fig 3B). The PERMANOVA of *Porites* corallum and corallite proximity matrix was significant among all comparisons ($p = 0.001$, S2 Table). Principal coordinate analysis (PCoA) of the proximity matrix showed three distinct clusters for each lineage, with minor overlapping. Corals were correctly classified (CC) despite clustering with specimens of different genetic lineages (Fig 4A). Among the 24 parameters included, theca thickness (THET), septa width (SEPW), lateral pali width (LPW) and space between septa (SEPS) were the most important variables, with the highest scores for MDA and MDG (Fig 4C and 5). For comparison, three clusters for each

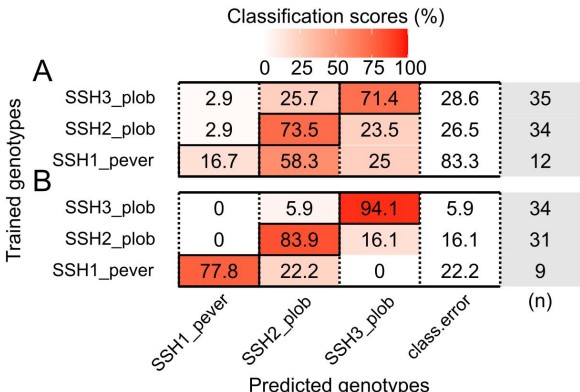

**Fig 3. *Porites* spp. confusion matrices of the Random Forest models.** (A) Corallum model – 35.08% out-of-bag error. (B) Corallum and corallite model – 12.16% out-of-bag error. Rows represent the genetic lineages used for training, and columns represent the predicted genetic lineages. The black outlined diagonal tiles indicate the correct classification scores. Class.error column indicates the classification error for each lineage. Numbers of individuals used for training each lineage are shown the column in grey.

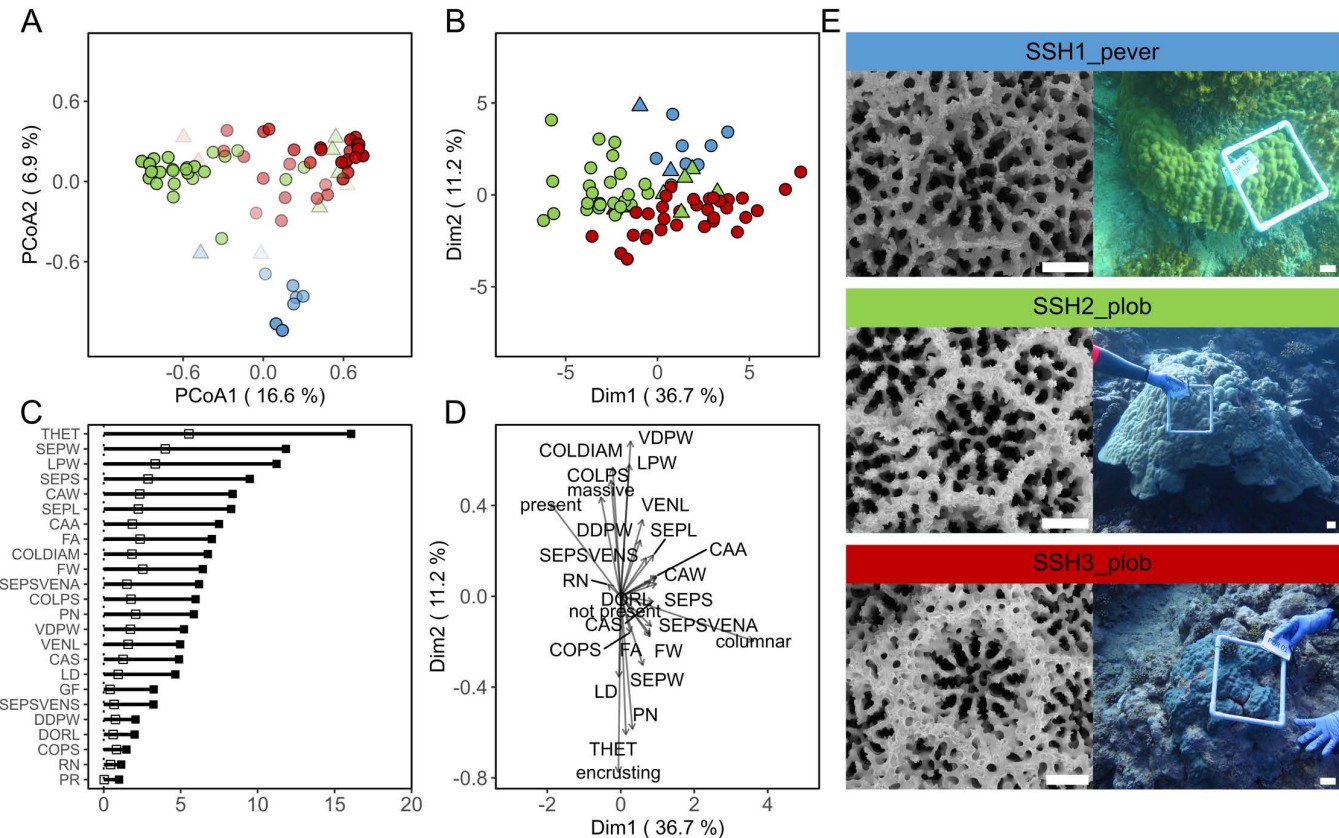

**Fig 4.** ***Porites* spp. corallum and corallite Random Forest model and Factor Analysis of Mixed Data.** (A) Principal coordinate analysis of *Porites* corallum and corallite model proximity matrix. (B) Principal coordinate analysis of factor analysis of mixed data (FAMD) on corallum and corallite morphological annotations. Colors indicate the different lineages (see below), and the shapes indicate whether RF correctly or incorrectly classified the sample (circle: CC, triangle: WC). Confidence of prediction is represented in A as an alpha gradient (0–1). (C) *Porites* corallum and corallite model variables of importance; mean decrease accuracy and mean decrease Gini are shown as black and white squares, respectively. (D) FAMD loadings of quantitative and qualitative parameters. (E) Corallite and corallum images of corals above 0.9 confidence of prediction by the RF model. SSH1_pever is represented in blue, SSH2_plob in green, SSH3_plob in red. Corallum and corallite image scale bars: 5 cm and 0.5 mm, respectively.

lineage were observed in the FAMD of *Porites* corallum and corallite morphology. The contribution of parameter loadings for the first two dimensions of the FAMD (Fig 4D) was consistent with the ones highlighted in the RF model. The distribution of CC and WC specimens according to the RF model seemed to be independent of the clustering defined by the FAMD morphospace (Fig 4B). A similar pattern was also observed in the PCoA on the Gower distances where CC and WC distribution was independent of the hierarchical clustering (S2B Fig). The pairwise PERMANOVA of the Gower distance matrix was significant between all comparisons ($p = 0.001$; S2 Table). Morphological variations in the corallum and corallites between genetic lineages are shown for colonies that received classification scores above 0.9 (Fig 4E).

### *Pocillopora* spp. models

The *Pocillopora* corallum model yielded an out-of-bag error of 59.68%. The highest accuracy scores were achieved for GSH09a at 73.1%, GSH09c at 61.5%, GSH13c at 50% (Fig 6A). The pairwise PERMANOVA on the *Pocillopora* corallum proximity matrix obtained by the RF model was significant among 47 out of the 78 comparisons ($p \leq 0.032$; S4 Table). Four major clusters were observed in the PCoA based on the RF proximity matrix, in which GSH13c and GSH09b were

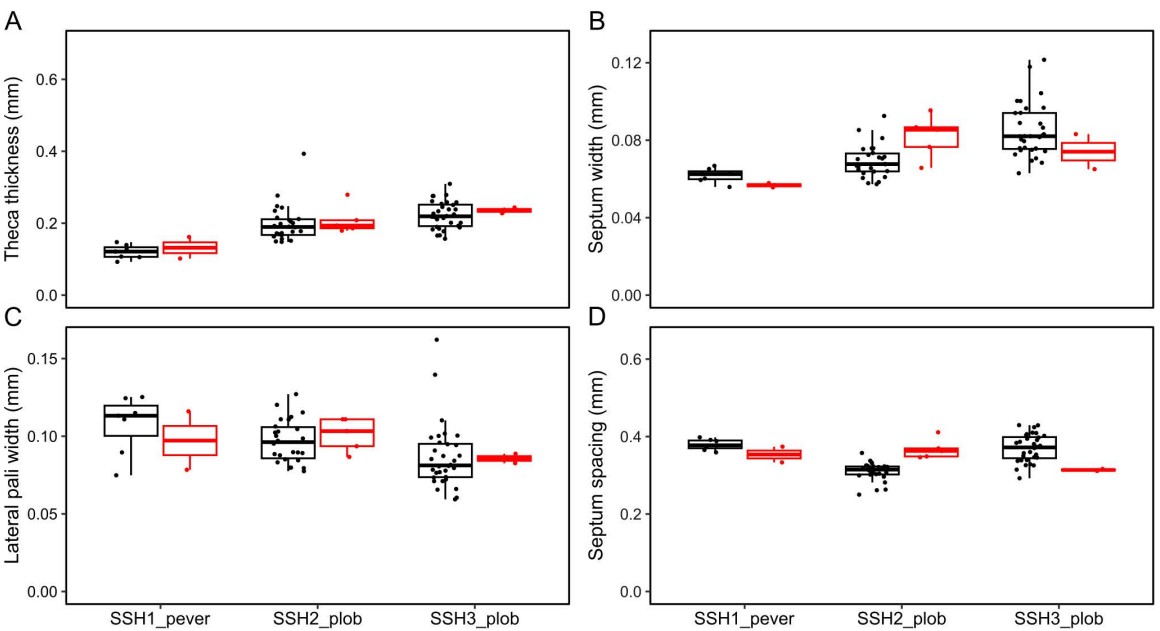

**Fig 5. The top four parameters derived from *Porites* spp. corallum and corallite model.** (A) Theca thickness (THET), (B) septum width (SEPW), (C) lateral pali width (LPW), and (D) septum spacing (SEPS). These parameters achieved the highest scores in mean decrease accuracy and mean decrease Gini.

divided into two clusters: (1) GSH13c, GSH09b, GSH014; (2) GSH13c, GSH09b, GSH13b; (3) GSH13a, GSH05, GSH15, GSH04; and (4) GSH09c, GSH01, GSH09a. Corals were correctly classified (CC) despite clustering with specimens of different genetic lineages (S4A Fig). Of the 14 parameters included, the five most relevant according to the MDA and MDG scores were branch spacing (BS), tip density (TD), verruca width (VW), verruca density (VD), and verruca length (VL) (S4C Fig). The principal component analysis and k-means clustering of corallum morphology showed multiple clusters overlapping and several lineages present in each cluster. The RF correctly classified (CC) specimens' distribution was independent of the k-means clustering (S4B Fig). No significance was found in the pairwise PERMANOVA applied on the *Pocillopora* euclidean distance matrix based on the corallum morphology (S4 Table).

*Pocillopora* corallum and corallite models achieved a 39.29% out-of-bag error, with the highest classification accuracies of 85.7% for GSH09c, 80% for GSH01, and 76% for GSH09a (Fig 6B). The pairwise PERMANOVA on the proximity matrix was significant between 56 out of the 78 comparisons ($p \leq 0.046$, S5 Table). The PCoA of the proximity matrix revealed three major clusters in the first two dimensions of the PCoA: (1) GSH09c, GSH01, and GSH15; (2) GSH09a, GSH12, GSH13b; (3) GSH09b, GSH13c, GSH13a, GSH05, GSH04, GSH10. Corals were correctly classified (CC) despite clustering with specimens of different genetic lineages (Fig 7A). Of the 27 parameters included, the four most relevant according to their MDA and MDG scores were the number of branch spacing (BS), septum width (SEPW), stylae number (STN), and mean of the four columella diameters (CODIAM) (Fig 7C and 8). Similar to the PCoA based on the RF proximity matrix, GSH09c, GSH01, and GSH15 were mostly separated from the other genetic lineages in the PCA of *Pocillopora* corallum and corallite morphology based on euclidean distance. The parameter loadings of STN, spinule width and spacing (SPIW, SPIS) and septa length (SEPL) contribute to GSH09c, GSH01, and GSH15 clustering on the first two dimensions of the PCA. The distribution of CC and WC specimens according to the RF model seemed to be independent of clustering in the PCA morphospace (Fig 7B). The k-means clustering observed in the PCA of corallum and corallite morphology showed multiple clusters overlapping and several lineages present in each cluster. The RF correctly classified (CC) specimens'

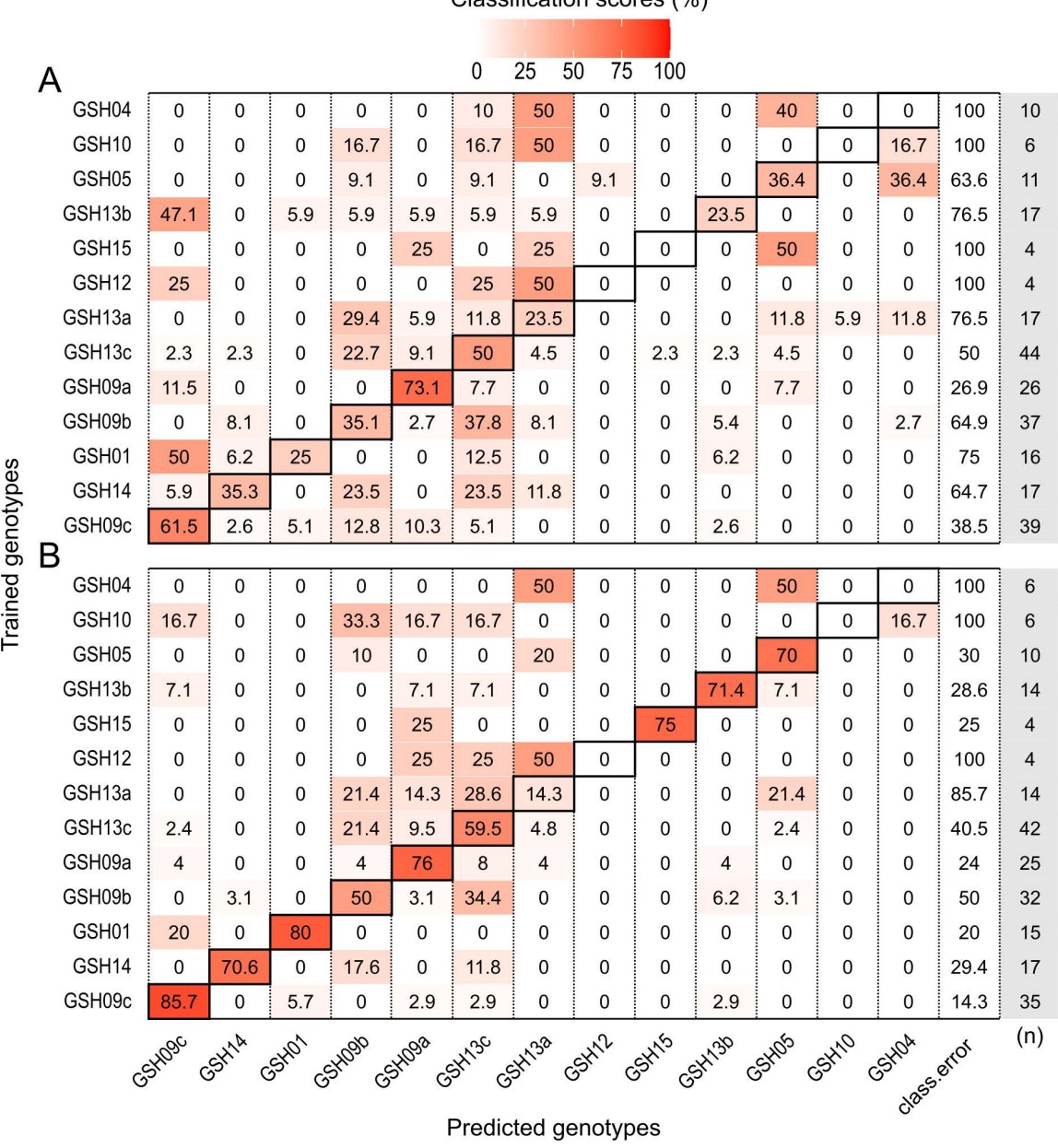

**Fig 6. *Pocillopora* spp. confusion matrices of the Random Forest models.** (A) Corallum model – 59.68% out-of-bag error. (B) Corallum and corallite model – 39.29% out-of-bag error. Rows represent the genetic lineage used for training, and columns represent the predicted genetic lineage. The black outlined diagonal tiles indicate the correct classification scores. Class.error column indicates the classification error for each lineage. Numbers of individuals used for training each lineage are shown the column in grey.

distribution was independent of the k-means clustering (Fig 7B). No significance was found in the pairwise PERMANOVA applied on the *Pocillopora* euclidean distance matrix based on the corallum and corallite morphology (S5 Table). Morphological variation between species can be observed on selected colony and corallite images that received classification scores above 0.9 (Fig 7E).

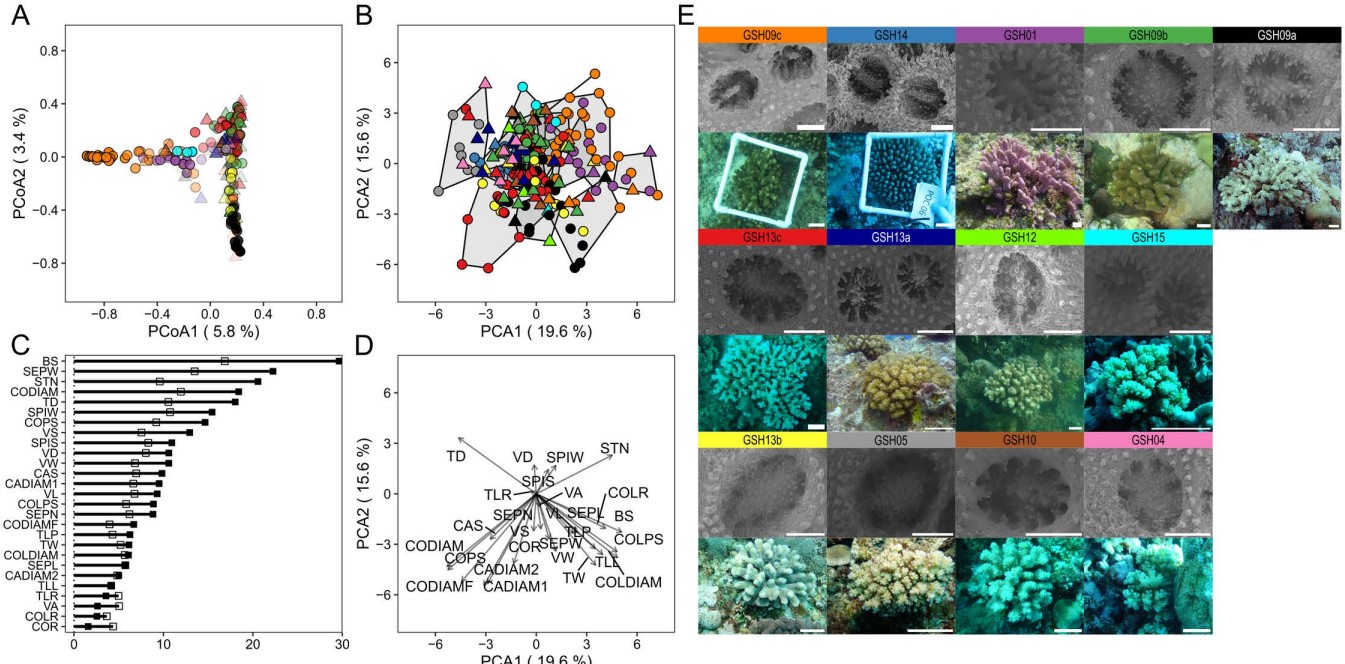

**Fig 7. *Pocillopora* spp. corallum and corallite Random Forest model and Principal component analysis.** (A) Principal coordinate analysis of *Pocillopora* corallum and corallite model proximity matrix. (B) Principal component analysis (PCA) on corallum and corallite morphological annotations. Colors indicate the different lineages (see below) and the shapes indicate whether the RF model correctly or incorrectly classified the samples (circle: CC, triangle: WC). Polygons in (B) show k-means clusters (k = 13). Confidence of prediction is represented in A as alpha gradient (0–1). (C) *Pocillopora* corallum and corallite model's variables of importance; mean decrease accuracy and mean decrease Gini are shown as black and white squares, respectively. (D) PCA parameter loadings. (E) Corallite and corallum images of corals above 0.9 confidence of prediction by the RF model. GSH09c represented in orange, GSH14 in blue, GSH01 in purple, GSH09b in green, GSH09a in black, GSH13c in red, GSH13a in dark blue, GSH12 in light green, GSH13b in yellow, GSH15 in cyan, GSH05 in grey, GSH10 in brown, GSH04 in pink. Corallum and corallite image scale bars: 5 cm and 0.5 mm, respectively.

## Discussion

Applying multiple lines of evidence to taxonomy arguably defines a robust species delimitation hypothesis [52,53]. Integrating genomics, morphology, the microbiome, and reproductive traits into coral systematics can help in resolving the species conundrum in corals [34,54,55]. To the best of our knowledge, this is the first study to classify and predict coral species using manual annotations of corallum and corallite morphology, assisted by a machine-learning-based algorithm. This was achieved in two widespread coral genera [56] collected from a vast geographical range across the Pacific and Indo-Pacific oceans.

*Porites* species identification based on both gross- and fine-scale morphologies is challenging. Colony growth can vary among encrusting, massive, columnar, plating, and branching forms, with some species presenting more than one [57,58]. Despite the 'simplistic' massive and encrusting shapes, which offer a limited number of measurable morphological parameters, RF-based analysis proved to be more efficient in resolving species limits compared to the traditionally used FAMD and Gower distances followed by hierarchical clustering. The parameter growth form (GF) proved to be the most important for the classification of the three species considered here. The distinguishing columnar shape of SSH3_plob likely contributed to the highly accurate predictions of SSH2_plob and SSH3_plob, whereas nearly 60% of SSH1_pever corals were incorrectly assigned as SSH2_plob possibly because of their overlapping growth forms (encrusting, massive) – see S3U Fig. When combining the corallum and corallite morphologies, both the RF proximity matrix and Gower distance could

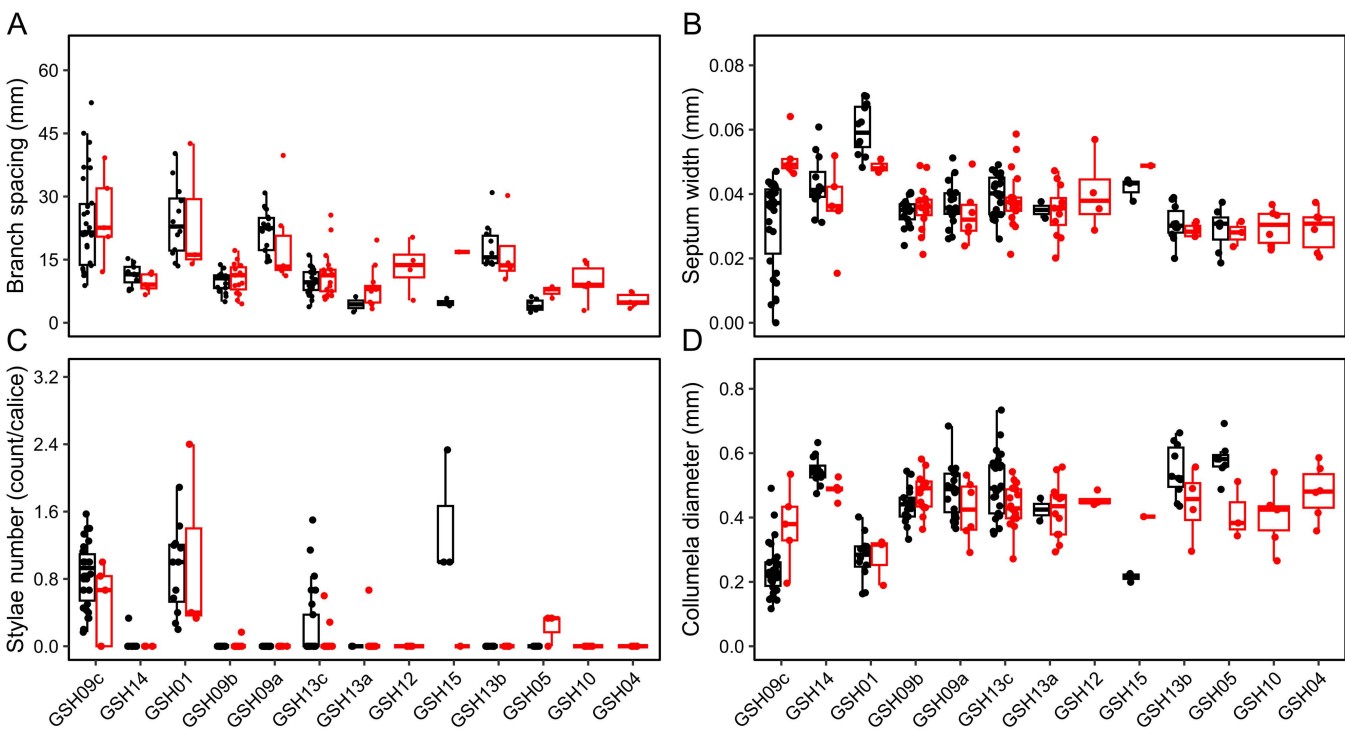

**Fig 8. The top four parameters derived from *Pocillopora* spp. corallum and corallite model.** (A) Branching spacing (BS), (B) septum width (SEPW), (C) stylae number (STN), and (D) columella diameter (CODIAM). These parameters achieved the highest scores in mean decrease accuracy and mean decrease Gini.

distinguish the three *Porites* species. *Porites lobata*, here a species complex including both SSH2_plob and SSH3_plob, and SSH1_pever were previously distinguished by combining molecular tools and morphometric measurements of the corallite in Forsman et al. [23]. Similarly, the RF model also highlighted the importance of corallite morphology (calice width (CAW), septa length (SEPL), and septa spacing (SEPS)) that significantly contributed to the accuracy of the model. *Porites* corals were sampled based on the massive *P. lobata* morphotype. The presence of a taxonomy expert underwater at the time of sampling was not always possible. As a result, only three morphologically similar genetic lineages were included in the *Porites* data set. The addition of more species and morphotypes, from a wider range of geographical locations and environmental conditions should be considered to ascertain robustness of the derived findings. Nonetheless, the models showed promising results to resolve two cryptic *P. lobata* genetic lineages (SSH2_plob and SSH3_plob) within *P. lobata* species, which were also found in sympatry [5], since they were successfully classified to their corresponding genetic lineages by the RF models. Our results support the morphological discrimination between these three species and highlight the effectiveness of species prediction based only on morphology using RF models.

Despite extensive studies on the phenotypic plasticity of *Pocillopora* [24,33,59], the morphological distinction between several species in the field remains challenging [6,31,32]. The complex shape of the branching *Pocillopora* genus allowed for the measurements of multiple traits, similar to previous studies [25,34]. The RF model proximities successfully resolved three out of the thirteen genetic lineages based on corallum morphological traits, and most of the lineages (nine out of the thirteen) when corallite morphology was added. In comparison, the traditional euclidean distances and k-means clustering approaches failed to distinguish genetic lineages when using either the corallum and corallite morphological traits. Machine-learning models can decrease in accuracy if the condition of the "large p, small n" paradigm is met in high dimensional data [60]. This

paradigm is exacerbated in the corallum model which also suffer from the low number of parameters measured as exemplified by the low accuracies for the classification of GSH12, GSH15, GSH10, and GSH04 – lineages with only a few individual replicates. Therefore, model accuracies could be improved by increasing the number of replicates per species. The columella diameter and the number of stylae on the columella were among the most important classification parameters. This agrees with previous studies that identified the columella type qualitatively as a distinctive trait of *Pocillopora* species [34], and quantitatively [6] the styliform columella, a distinctive feature for GSH09c lineage currently described as *P. grandis*, which also shows to be a prominent characteristic for GSH01 lineage, described as *P.* cf. *effusa* (Fig 8). Overall, these results suggest that morphological identification of *Pocillopora* and *Porites* species is achievable through a machine-learning algorithm and by using reproducible quantitative measurements at the corallum and corallite skeletal levels.

Dimension reduction analysis, such as PCA and FAMD, are unsupervised clustering methods that can be used to investigate clustering patterns in multivariate data sets while retaining most of the variation [47,61]. In a taxonomic context, these analyses are commonly applied to identify possible morphological groupings based on variations in multiple traits that can be visualized in a biplot. Hierarchical and K-means clustering are also widely used algorithms in taxonomy [62]. These are also unsupervised methods where a predefined number of clusters should be defined prior to analysis. RF models can also be used in the same manner, with the advantages of handling mixed and missing data and being insensitive to linear scaling and outliers [63]. In a previous study, unsupervised RF modelling followed by hierarchical clustering was used to identify corallum morphotypes associated with different genetic lineages of *Pocillopora* and *Porites* corals [5]. Here, supervised RF models were further able to predict the correct genetic lineages based on morphology, even for cryptic species (e.g., SSH2_plob and SSH3_plob; GSH014 now described as *P. tuahiniensis*) that shared the same morphological clusters in the PCoA generated from the proximity matrix of the RF models, and in the dimension reduction and clustering analyses. Furthermore, the RF models performed better than the Euclidean- and Gower-based distance approaches in morphologically delimiting the species in the two genera tested. These findings highlight that in modern taxonomy studies where genome-wide sequencing strategies are resolutive approaches for genetic lineages delimitations [64], morphology trait analysis can be improved to assist integrative taxonomy. Moreover, RF models have been shown to successfully solve classification problems in other scientific fields [11,17,65] and could be used as an alternative classification method for corals, where it not only allows prediction, but also identifies the morphological traits of importance, similar to the conventional methods used in coral systematics.

Advances in deep-learning approaches have shown promising results for species-level identification based on in-situ images [26, 66] which can be applied to large-scale non-destructive ecological surveys. Computer vision combined with neural networks were able to effectively emulate a specialist discernment to classify coral species based on micro morphology [67]. These algorithms require a large number of images and rely on the color, texture, and contour shape of the targeted organism for training and prediction. However, the measurements of morphological traits – which may be important delimiting factors – are not normally considered. This greatly limits further investigation of the phylogeny of coral species. The workflow proposed here, which combines manual morphological annotations with machine learning analysis, allows access to the importance of measured traits for the predictability of the model – MDA and MDG. These are considered similar to the contribution of parameter loadings in PCA or FAMD [63]. Therefore, in addition to their use in the classification of coral species using manual morphological annotations of field images, RF models also provide relevant morphological information that can be later assessed for further ecological and phylogenetic studies. Another feature of the RF model that is of interest to taxonomists and ecologists is the number of votes that a sample is given to each class label (e.g., genetic lineage) during prediction. This allows us to estimate the confidence level with which the model classified the sample. Even when a low accuracy is obtained, labels that receive the majority of votes can be used as identifiers of the closest genetic lineage in a given sample. Therefore, RF models can be used for primary species hypotheses in the absence of a specialist, using corallum morphology alone or in combination with corallite analysis for both field imaging and laboratory identification.

## Conclusion

Physiological and morphological plasticity, hybridization, convergence, and geographical isolation are among the factors that contribute to morphological variation in hermatypic corals. These variables and unpredictable changes can hinder the congruency between genomic and morphological analyses. In the era of comparative genomics, morphology is sometimes overlooked [68]. Morphological studies are key to determining the physical interactions between an organism, the environment, and other organisms. Our results demonstrate the capability of a machine-learning algorithm to classify and predict coral species using the manual annotation of coral skeletons. Four models were applied to two genera known to present overlapping morphological traits among species. Despite the high predictive accuracies achieved by the models, this study could not determine whether the observed variation is genetically or environmentally driven, due to sampling constraints, and the topic should be further explored in future studies. Notably, the measurements conducted are reproducible without the explicit need for taxonomic expertise and can be achieved with a diving camera and computer, making it a cost-effective and accessible method even for citizen scientists. The potential of using machine learning to delineate coral species based on morphology could be applied in phylogenetic and systematic studies, in the absence of genetic material, such as paleo ecological studies relying on fossils. In conclusion, the method presented here should be considered in an integrative taxonomy workflow.

## Supporting information

**S1 File. S1 Appendix.**
(DOCX)

## Acknowledgments

Special thanks to the Tara Ocean Foundation, the R/V Tara crew, and the Tara Pacific Expedition Participants (https://doi.org/10.5281/zenodo.3777760). The authors also particularly thank Serge Planes, Denis Allemand, and the Tara Pacific consortium, Tara Pacific Foundation teams. We also thank all the Tara Pacific Coordinators: Serge Planes (project director and lead author, planes@univ-perp.fr), PSL Research University: EPHE-UPVD-CNRS, USR 3278 CRIOBE, Laboratoire d'Excellence CORAIL, Université de Perpignan, 52 Avenue Paul Alduy, 66860 Perpignan Cedex, France; Stéphanie Reynaud, Centre Scientifique de Monaco, 8 Quai Antoine Ier, MC-98000, Principality of Monaco; Denis Allemand (project director and lead author, allemand@centrescientifique.mc), Centre Scientifique de Monaco, 8 Quai Antoine Ier, MC-98000, Principality of Monaco; Bernard Banaigs, PSL Research University: EPHE-UPVD-CNRS, USR 3278 CRIOBE, Université de Perpignan, France;Sylvain Agostini, Shimoda Marine Research Center, University of Tsukuba, 5-10-1, Shimoda, Shizuoka, Japan; Emilie Boissin PSL Research University: EPHE-UPVD-CNRS, USR 3278 CRIOBE, Laboratoire d'Excellence CORAIL, Université de Perpignan, 52 Avenue Paul Alduy, 66860 Perpignan Cedex, France; Emmanuel Boss School of Marine Sciences, University of Maine, Orono, 04469, Maine, USA; Chris Bowler,Institut de Biologie de l'Ecole Normale Supérieure (IBENS), Ecole normale supérieure, CNRS, INSERM, Université PSL, 75005 Paris, France; Colomban de Vargas Sorbonne Université, CNRS, Station Biologique de Roscoff, AD2M, UMR 7144, ECOMAP 29680 Roscoff, France & Research Federation for the study of Global Ocean Systems Ecology and Evolution, FR2022/ Tara Oceans-GOSEE, 3 rue Michel-Ange, 75016 Paris, France; Eric Douville, Laboratoire des Sciences du Climat et de l'Environnement, LSCE/IPSL, CEA-CNRS-UVSQ, Université Paris-Saclay, F-91191 Gif-sur-Yvette, France; Michel Flores, Weizmann Institute of Science, Department of Earth and Planetary Sciences, 76100 Rehovot, Israel; Didier Forcioli, LIA ROPSE, Laboratoire International Associé Université Côte d'Azur—Centre Scientifique de Monaco, Monaco, Principality of Monaco; Paola Furla, Université Côte d'Azur, CNRS, INSERM, IRCAN, Medical School, Nice, France and Department of Medical Genetics, CHU of Nice, France; Pierre Galand, Sorbonne Université, CNRS, Laboratoire d'Ecogéochimie des Environnements Benthiques (LECOB), Observatoire Océanologique de Banyuls, 66650 Banyuls sur mer, France; Eric

Gilson, Université Côte d'Azur, CNRS, Inserm, IRCAN, France; Fabien Lombard, Sorbonne Université, Institut de la Mer de Villefranche sur mer, Laboratoire d'Océanographie de Villefranche, F-06230 Villefranche-sur-Mer, France; Stéphane Pesant, European Molecular Biology Laboratory, European Bioinformatics Institute, Wellcome Genome Campus, Hinxton, Cambridge CB10 1SD, UK; Matthew B. Sullivan, The Ohio State University, Departments of Microbiology and Civil, Environmental and Geodetic Engineering, Columbus, Ohio, 43210 USA; Shinichi Sunagawa, Department of Biology, Institute of Microbiology and Swiss Institute of Bioinformatics, Vladimir-Prelog-Weg 4, ETH Zürich, CH-8093 Zürich, Switzerland; Olivier Thomas, Marine Biodiscovery Laboratory, School of Chemistry and Ryan Institute, National University of Ireland, Galway, Ireland; Romain Troublé, Fondation Tara Océan, Base Tara, 8 rue de Prague, 75 012 Paris, France; Rebecca Vega Thurber, Oregon State University, Department of Microbiology, 220 Nash Hall, 97331Corvallis OR USA; Christian R. Voolstra, Department of Biology, University of Konstanz, 78457 Konstanz, Germany; Patrick Wincker, Génomique Métabolique, Genoscope, Institut François Jacob, CEA, CNRS, Univ Evry, Université Paris-Saclay, 91057 Evry, France; Maren Ziegler, Marine Holobiomics Lab, Department of Animal Ecology and Systematics, Justus Liebig University Giessen, Giessen, Germany; Didier Zoccola, Centre Scientifique de Monaco, 8 Quai Antoine Ier,MC-98000, Principality of Monaco. We also would like to thank the SequAna Core lab at the University of Konstanz for their help with data curation and analysis (BCCH). The Tara Pacific expedition would not have been possible without the participation and commitment of over 200 scientists, sailors, artists and citizens (see https://zenodo.org/record/3777760#.YfEEsfXMLjB). Also, thanks to Professor Yokochi Hiroyuki for assisting on the decision of key morphological parameters to be measured. This publication is number 42 of theTara Pacific Consortium.

## Author contributions

**Conceptualization:** Guinther Mitushasi.

**Data curation:** Guinther Mitushasi, Yuko F Kitano, Nicolas Oury, David A. Paz-García, Sylvain Agostini.

**Formal analysis:** Guinther Mitushasi.

**Funding acquisition:** Guinther Mitushasi, Serge Planes, Denis Allemand, Sylvain Agostini.

**Investigation:** Guinther Mitushasi.

**Methodology:** Guinther Mitushasi, Sylvain Agostini.

**Resources:** Nicolas Oury, Hélène Magalon, David A. Paz-García, Eric Armstrong, Benjamin C. C. Hume, Barbara Porro, Clémentine Moulin, Emilie Boissin, Guillaume Bourdin, Guillaume Iwankow, Julie Poulain, Sarah Romac, Serge Planes, Denis Allemand, Christian R Voolstra, Didier Forcioli, Sylvain Agostini.

**Software:** Guinther Mitushasi, Sylvain Agostini.

**Supervision:** Sylvain Agostini.

**Visualization:** Guinther Mitushasi.

**Writing – original draft:** Guinther Mitushasi.

**Writing – review & editing:** Guinther Mitushasi, Yuko F Kitano, Nicolas Oury, Hélène Magalon, David A. Paz-García, Eric Armstrong, Benjamin C. C. Hume, Barbara Porro, Clémentine Moulin, Emilie Boissin, Guillaume Bourdin, Guillaume Iwankow, Julie Poulain, Sarah Romac, Maggie M. Reddy, Serge Planes, Denis Allemand, Christian R Voolstra, Didier Forcioli, Sylvain Agostini.

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
