## [Decision Letter · Decision Letter 0]

May 31 2025

Dear Dr. Mitushasi,

Thank you for submitting your manuscript to PLOS ONE. After careful consideration, we feel that it has merit but does not fully meet PLOS ONE’s publication criteria as it currently stands. Therefore, we invite you to submit a revised version of the manuscript that addresses the points raised during the review process.

We look forward to receiving your revised manuscript.

Kind regards,

Shashank Keshavmurthy, PhD

Academic Editor

PLOS ONE

**Journal Requirements:**

1. When submitting your revision, we need you to address these additional requirements. Please ensure that your manuscript meets PLOS ONE's style requirements, including those for file naming. The PLOS ONE style templates can be found at https://journals.plos.org/plosone/s/file?id=wjVg/PLOSOne_formatting_sample_main_body.pdf and https://journals.plos.org/plosone/s/file?id=ba62/PLOSOne_formatting_sample_title_authors_affiliations.pdf 2. Thank you for stating in your Funding Statement: Work from GM is supported by JST SPRING (https://www.jst.go.jp/jisedai/spring/en/outline/index.html), Grant Number JPMJSP2124. The funders had no role in study design, data collection and analysis, decision to publish, or preparation of the manuscript. Please provide an amended statement that declares *all* the funding or sources of support (whether external or internal to your organization) received during this study, as detailed online in our guide for authors at http://journals.plos.org/plosone/s/submit-now.  Please also include the statement “There was no additional external funding received for this study.” in your updated Funding Statement. Please include your amended Funding Statement within your cover letter. We will change the online submission form on your behalf. 3. Thank you for stating the following in the Acknowledgments Section of your manuscript: Special thanks to the Tara Ocean Foundation, the R/V Tara crew and the Tara Pacific Expedition Participants (https://doi.org/10.5281/zenodo.3777760). We are keen to thank the commitment of the following institutions for their financial and scientific support that made this unique Tara Pacific Expedition possible: CNRS, PSL, CSM, EPHE, Genoscope, CEA, Inserm, Université Côte d’Azur, ANR, agnès b., UNESCO-IOC, the Veolia Foundation, the Prince Albert II de Monaco Foundation, Région Bretagne, Billerudkorsnas, AmerisourceBergen Company, Lorient Agglomération, Oceans by Disney, L’Oréal, Biotherm, France Collectivités, Fonds Français pour l’Environnement Mondial (FFEM), Etienne Bourgois, and the Tara Ocean Foundation teams. Tara Pacific would not exist without the continuous support of the participating institutes. The authors also particularly thank Serge Planes, Denis Allemand, and the Tara Pacific consortium. This is publication number #00 of theTara Pacific Consortium. We would like to thank the SequAna Core lab at the University of Konstanz for their help with data curation and analysis (BCCH). Also, thanks to Professor Yokochi Hiroyuki for assisting on the decision of key morphological parameters to be measured. We note that you have provided funding information that is not currently declared in your Funding Statement. However, funding information should not appear in the Acknowledgments section or other areas of your manuscript. We will only publish funding information present in the Funding Statement section of the online submission form. Please remove any funding-related text from the manuscript and let us know how you would like to update your Funding Statement. Currently, your Funding Statement reads as follows: Work from GM is supported by JST SPRING (https://www.jst.go.jp/jisedai/spring/en/outline/index.html), Grant Number JPMJSP2124. The funders had no role in study design, data collection and analysis, decision to publish, or preparation of the manuscript. Please include your amended statements within your cover letter; we will change the online submission form on your behalf. 4. When completing the data availability statement of the submission form, you indicated that you will make your data available on acceptance. We strongly recommend all authors decide on a data sharing plan before acceptance, as the process can be lengthy and hold up publication timelines. Please note that, though access restrictions are acceptable now, your entire data will need to be made freely accessible if your manuscript is accepted for publication. This policy applies to all data except where public deposition would breach compliance with the protocol approved by your research ethics board. If you are unable to adhere to our open data policy, please kindly revise your statement to explain your reasoning and we will seek the editor's input on an exemption. Please be assured that, once you have provided your new statement, the assessment of your exemption will not hold up the peer review process. 5. One of the noted authors is a group or consortium Tara Pacific Consortium. In addition to naming the author group, please list the individual authors and affiliations within this group in the acknowledgments section of your manuscript. Please also indicate clearly a lead author for this group along with a contact email address. 6. Please include your full ethics statement in the ‘Methods’ section of your manuscript file. In your statement, please include the full name of the IRB or ethics committee who approved or waived your study, as well as whether or not you obtained informed written or verbal consent. If consent was waived for your study, please include this information in your statement as well.

**Additional Editor Comments:**

Dear Authors

I have with me response from 1 reviewer, It was very difficult to find even one reviewer to read this manuscript. As you can see, the reviewer has pointed out certain aspects of the manuscript that needs to be addressed, technical and general

please go through it and revise the manuscript

best of luck

sincerely

Reviewers' comments:

Reviewer's Responses to Questions

**Comments to the Author**

1. Is the manuscript technically sound, and do the data support the conclusions?

Reviewer #1: Yes

2. Has the statistical analysis been performed appropriately and rigorously?

Reviewer #1: I Don't Know

3. Have the authors made all data underlying the findings in their manuscript fully available?

Reviewer #1: Yes

4. Is the manuscript presented in an intelligible fashion and written in standard English?

Reviewer #1: Yes

**Reviewer #1:**  I’ve read the manuscript by Guinther Mitushasi and collaborators. The authors used Random Forest models combining micro and macro morphological measurements of 2 coral genera (Porites and Pocillopora) and verified that these models can better predict genetic lineage than more traditional methods such as PCA and FAMD.

I think the manuscript is well-written and the large dataset used here strongly supporting the authors’ results and discussion. I only have a series of small but important comments that I think could improve the manuscript.

Comment #1

First of all, I have great respect for the previous work of Gélin et al. and Voolstra et al., but species name and genotype name should not be used as synonyms. To follow the manuscript title, the authors could just focus on genetic lineages throughout the manuscript and keep the species assignment for the discussion section.

I think this should be done for clarity as well.

Comment #2

L70-71: I would tone down this sentence as we still need some sort of taxonomic expertise/reference… at least to tell our machine what to measure and how to measure it.

Just picking the first google reference mentioning this issue (I’m not a co-author on it): https://link.springer.com/article/10.1007/s10531-024-02934-6.

From my point of view the word ‘eliminating’ is way too strong. Maybe ‘reducing’?

Comment #3

L71-72: Since your best model predictor includes corallum and corallite data I would just write ‘support image identification’. Somehow, we still need to collect a small piece of coral to identify our specimens more accurately, right?

Comment #4

L89-90: Clustering methods are specifically called here, and I was hoping that it would also be tested later… k-means in particular is being used as well to delineate species… Are you considering comparing Random Forest vs clustering methods in the near future?

Comment #5

L115: Correct me if I’m wrong but the method presented here still requires diving, sampling, and a substantial amount of processing (bleaching, SEM, etc), right? If the method used here provides a better prediction of genotype/species identify using micromorphology, then collection/destructive methods cannot be avoided.

Comment #6

L139-148: Related to my comment #1 and if the authors are focusing on genotypes, please provide a genotype name for P. evermanni, indicate between brackets to what species GSH12 and 15 might correspond to (even a cf., aff., or undescribed sp would do).

Comment #7

L164-165: There is an evident question here: Can the observed variation in this manuscript be explain by the geographical origin of the specimens?

Comment #8

L166: Sorry, I couldn’t resist emphasizing that this is the reason why we still need some sort of taxonomic expertise when we collect specimens in the field. Despite focusing on Porites lobata morphology, none of the specimens were eventually genotyped as P. lobata.

Comment #9

L171: I know that identifying Pocillopora species from the field is particularly difficult, so I can only say that adding more specimens here is definitely giving strength to the authors’ methods and results! However, now I feel like the authors should probably discuss (discussion section) the limitation of effectively identifying Porites genotype if more specimens from different species were added.

Comment #10

L202: About the morphological measurements, I could not find how many replicates were done for the corallum measurements. This could be added in the Table 1 and 2 or in the S1 Appendix as well.

Actually, it would have been great if those very beautiful Figure S5, 6, 7, and 8 could be combine with Tables 1 and 2 (which by the way really look like the Table S8 and 9). I would like to know what the editor is thinking about that. The paper would become visually cool!

Comment #11

L229: Give a genotype name to P. evermanni and comment #1 would be fixed. I would have more suggestions for you later to arrange the figure accordingly.

Comment #12

L274, 295, 332: Figures 1 and 4 are so hard to read as they are. If the reader is not used to confusion matrices he doesn’t know what the ligns/columns correspond to. Genotype X (line) effectively assigned to genotype X (column)?

In addition, I would suggest providing a genotype name for P. evermanni.

For the column either use genotype names OR species names but not at the same time. It’s okay to provide a genotype name to specimen (P. evermannii) or a species name to specimen that are not officially named (Porites sp1 and Porites sp2).

I would have emphasized the genotypes and only discuss species in the discussion section.

Comment #13

L289, Appendix Tables: When reporting R2 and F values we really don’t need 8 digits. I think 2 are enough. For p and p.adj values, the authors can also use a generalized p<0.05, p<0.01, p<0.001 or if they prefer to give an exact value please make sure the number of digits is homogene throughout the manuscript (main text and tables).

Comment #14

Sorry to emphasize this again but either discuss species OR genotype and not both at the same time. The discussion used species and genotypes as if there were synonyms and this is not correct. The authors could provide a genotype name to P. evermanni for an easy fix.

Alternatively, this would be the place to shift the discussion from genotype to species by assigning each genotype to a tentative species name (Porites evermanni, Porites cf/aff/X or Porites sp1, etc.), and use the species name throughout the manuscript.

L420: ‘The two cryptic P.lobata species (SSH2_plob and SSH3_plob)’ is a perfect example of what should be avoided.

‘The cryptic species P. lobata, which includes 2 genetic variants SSH2_plob and SSH3_plob’ would be more correct. There is probably more ways to turn this sentence.

Comment #15

L424: I would also add that the findings for Porites should be further tested to include more specimens and morphologies.

L448: I haven’t seen any discussion on the potential bias that this study could have produced. For example, were specimens always collected from shallow, protected reefs? Or were they collected from various locations encompassing large environmental gradients? Also, the sampling design was targeting 1 particular morphology/morphotype/species, right? If this issue was probably solved for Pocillopora, I think the authors should add a few lines on this potential bias for Porites.

Comment #16

L451: Double reference.

Comment #17

L460-461: A bit of rewording is required here to avoid using genotypes and species names as synonyms.

Comment #18

L473-477: Generally speaking, I’m a little confused about how the authors connect deep-learning approaches, in-situ species ID, and micro-morphology (corallites?). Instead of “in-situ” I would emphasize that their method can allow image identification of coral species.

**Do you want your identity to be public for this peer review?** For information about this choice, including consent withdrawal, please see our Privacy Policy

Reviewer #1: No

---

## [Author Response · Author response to Decision Letter 1]

1 May 2025

Dear Editor and Reviewer,

We have addressed all the corrections proposed in the comments by the reviewer. It includes streamlining the genotype naming instead of using species names interchangeably, the merging of figures and tables containing the morphological measurements descriptions from the supplementary to the main text, and the addition of two clustering analysis for comparison with Random Forest. We have also provided the necessary edits for the manuscript, and related files to meet the requirements and standards of the journal. Additionally, all the related data and analysis related to the manuscript were made available at a public access repository.

We hope the answers and modifications provided can satisfy the reviewers’ and editor’s standards for publication.

Reviewer #1: I’ve read the manuscript by Guinther Mitushasi and collaborators. The authors used Random Forest models combining micro and macro morphological measurements of 2 coral genera (Porites and Pocillopora) and verified that these models can better predict genetic lineage than more traditional methods such as PCA and FAMD.

I think the manuscript is well-written and the large dataset used here strongly supporting the authors’ results and discussion. I only have a series of small but important comments that I think could improve the manuscript.

We would first thank the reviewer for his/her careful and constructive evaluation of our manuscript. We sincerely appreciate the insightful comments, which have significantly contributed to improving the clarity and overall quality of the work. Please see below for the answers to the comments.

Comment #1

First of all, I have great respect for the previous work of Gélin et al. and Voolstra et al., but species name and genotype name should not be used as synonyms. To follow the manuscript title, the authors could just focus on genetic lineages throughout the manuscript and keep the species assignment for the discussion section.

I think this should be done for clarity as well.

We thank the Reviewer for highlighting the inconsistency in using genetic lineages and species names interchangeably. We have revised the manuscript to ensure consistent use of genetic lineage terminology throughout the manuscript. Specific changes and clarifications are detailed in our responses to the relevant comments below.

Comment #2

L70-71: I would tone down this sentence as we still need some sort of taxonomic expertise/reference… at least to tell our machine what to measure and how to measure it.

Just picking the first google reference mentioning this issue (I’m not a co-author on it): https://link.springer.com/article/10.1007/s10531-024-02934-6.

From my point of view the word ‘eliminating’ is way too strong. Maybe ‘reducing’?

Yes, thank you, this is indeed a very important distinction: we replaced the word ‘eliminating’ to ‘reducing’ (L73).

Comment #3

L71-72: Since your best model predictor includes corallum and corallite data I would just write ‘support image identification’. Somehow, we still need to collect a small piece of coral to identify our specimens more accurately, right?

Thank you for pointing out this important clarification. We changed from ‘in field identification’ to ‘image identification’(L74).

Comment #4

L89-90: Clustering methods are specifically called here, and I was hoping that it would also be tested later… k-means in particular is being used as well to delineate species… Are you considering comparing Random Forest vs clustering methods in the near future?

Thank you for proposing the comparison between Random Forest and clustering methods. We addressed this by including two clustering methods for comparison in the manuscript. We used hierarchical clustering for Porites using the distance matrix acquired from the Gower distances, since these data have both qualitative and quantitative parameters. This was plotted in a principal coordinate analysis for both the corallum alone and corallum/corallite morphologies (S1 Appendix, S2 Fig). For Pocillopora, we indeed used k-means on the corallum alone and corallum/corallite morphologies. These were plotted on the already existent PCAs (Fig 7 and S4 Fig). Additional information related to the clustering analysis can be found in the revised Materials and Methods (L279 and from L283 to L292), Results (L321 to 324, L348 to 350, L387 to 390, L418 to 421), and Discussion (L459, L492, L513 to 516).

Comment #5

L115: Correct me if I’m wrong but the method presented here still requires diving, sampling, and a substantial amount of processing (bleaching, SEM, etc), right? If the method used here provides a better prediction of genotype/species identify using micromorphology, then collection/destructive methods cannot be avoided.

Thank you for your comment. Yes, you are correct that the micromorphology method is still invasive, since it requires sampling of the colony skeleton. However, the method can be non-invasive if solely macromorphology is measured on in-situ images of the colony. Applying the model based on the corallum was specifically designed for this purpose. We have added the word “noninvasive” to L119. “Linear morphometrics were applied to the corallum and corallite morphologies of two widespread coral genera (Porites and Pocillopora) to generate (1) a model for noninvasive in-situ image identification of coral species using corallum morphology, and (2) a model for species identification using both corallite and corallum morphological traits.”

The accuracies are lower than the combined macro- and micromorphology model. However, it still allows identification without sampling and with minimal prior taxonomic knowledge. As discussed in the revised manuscript (L212-214), the in-situ image dataset was limited to one image per sample, but we believe the addition of more parameters would likely increase the accuracy of the model.

Comment #6

L139-148: Related to my comment #1 and if the authors are focusing on genotypes, please provide a genotype name for P. evermanni, indicate between brackets to what species GSH12 and 15 might correspond to (even a cf., aff., or undescribed sp would do).

We thank you for noticing this issue. We have provided a genotype name for P. evermanni (SSH1_pever).

(L142) “The Porites samples were assigned to three species: P. evermanni, hereafter referred to as SSH1_pever, and two new cryptic lineages within Porites lobata species not described in Hellberg et al.[35] referred to as SSH2_plob and SSH3_plob” [5].

We also added the tentative corresponding species names for GSH12, GSH15. Additionally, we added tentative species names to other genotypes, which were unnamed (GSH09a, GSH13a).

(L152 to 157) “According to Oury et al. [6] GSH09a correspond to a distinct species from P. meandrina referred to here as P aff. meandrina. GHS13a is a species undescribed or incorrectly synonymized with P. verrucosa, here referred to as P. aff. verrucosa. GSH12 is a hybrid between GSH13c and GSH13a here referred to as P. verrucosa x P aff. verrucosa. GSH15 is a hybrid between GSH13c and GSH14 referred to as P. verrucosa x P tuahiniensis.”

Comment #7

L164-165: There is an evident question here: Can the observed variation in this manuscript be explain by the geographical origin of the specimens?

Thank you. We agree with the evident question of whether morphological variation could be explained by geographical origin. For samples collected during the Tara Pacific expedition, an average of three colonies per site was used. This imposes a certain limit to investigate this question. It is true that according to Voolstra et al. (2023) the Pocillopora genotypes referred to as P. verrucosa and P. tuahiniensis, and P. meandrina and P. grandis are genetically similar species pairings and only found in allopatry in our samples. For Porites genotypes, SSH2_plob and SSH3_plob (the more genetically similar lineages) were found in widespread sympatry. In Oury et al. (2023), sampling was not always morphotype-based, and many species were found in sympatry. Here we argue that rather than geographical origin, the environmental conditions (including depth) at the site of origin should be considered, since it can affect the morphological variation through phenotypic plasticity. This is planned to be investigated in follow-up studies by including more samples and locations in the morphological data set. To address your comment, we have added the following sentence to the conclusion:

(L569 to 572)“Despite the high predictive accuracies achieved by the models, this study could not determine whether the observed variation is genetically or environmentally driven, due to sampling constraints, and the topic should be further explored in future studies.”

Comment #8

L166: Sorry, I couldn’t resist emphasizing that this is the reason why we still need some sort of taxonomic expertise when we collect specimens in the field. Despite focusing on Porites lobata morphology, none of the specimens were eventually genotyped as P. lobata.

Thank you for emphasizing the importance of involving a taxonomy expert during sampling. We fully agree that the presence of a trained taxonomist is ideal for field sampling. We hope that a future model may facilitate, and aid correct identification in the field.

Comment #9

L171: I know that identifying Pocillopora species from the field is particularly difficult, so I can only say that adding more specimens here is definitely giving strength to the authors’ methods and results! However, now I feel like the authors should probably discuss (discussion section) the limitation of effectively identifying Porites genotype if more specimens from different species were added.

We thank the reviewer for highlighting the lack of discussion about the limitations in the Porites models. We added these considerations to the Discussion section.

(L472 to 481) “Porites corals were sampled based on the massive P. lobata morphotype. The presence of a taxonomy expert underwater at the time of sampling was not always possible. As a result, only three morphologically similar genetic lineages were included in the Porites data set. The addition of more species and morphotypes, from a wider range of geographical locations and environmental conditions should be considered to ascertain robustness of the derived findings. Nonetheless, the models showed promising results to resolve two cryptic P. lobata genetic lineages (SSH2_plob and SSH3_plob) within P. lobata species, which were also found in sympatry [5], since they were successfully classified to their corresponding genetic lineages by the RF models.”

Comment #10

L202: About the morphological measurements, I could not find how many replicates were done for the corallum measurements. This could be added in the Table 1 and 2 or in the S1 Appendix as well.

Actually, it would have been great if those very beautiful Figure S5, 6, 7, and 8 could be combine with Tables 1 and 2 (which by the way really look like the Table S8 and 9). I would like to know what the editor is thinking about that. The paper would become visually cool!

We have edited a sentence to clarify the number of replicates conducted for the corallum measurements.

(L212-214) “Corallum measurements, were conducted on one whole colony image per sample [5], and one branch fragment image per sample in the case for Pocillopora fragments.”

Thank you for the suggestions on combining the morphological measurements description tables and figures. We combined Table 1 and 2 with Table S8 and S9. We have also combined Fig. S5 and 6 for Porites and Fig. S7 and 8 for Pocillopora and added the revised figures to the main text.

Comment #11

L229: Give a genotype name to P. evermanni and comment #1 would be fixed. I would have more suggestions for you later to arrange the figure accordingly.

We addressed this suggestion in the answer to comment #6

Comment #12

L274, 295, 332: Figures 1 and 4 are so hard to read as they are. If the reader is not used to confusion matrices he doesn’t know what the ligns/columns correspond to. Genotype X (line) effectively assigned to genotype X (column)?

In addition, I would suggest providing a genotype name for P. evermanni.

For the column either use genotype names OR species names but not at the same time. It’s okay to provide a genotype name to specimen (P. evermannii) or a species name to specimen that are not officially named (Porites sp1 and Porites sp2).

I would have emphasized the genotypes and only discuss species in the discussion section.

We thank you for the suggested changes to Fig 1 and 4, which are now Fig 3 and 6 in the revised manuscript. As elucidated in previous comments, we have added a genotype to P. evermanni (SSH1_pever). We also edited all instances of P. evermanni that were used together with genotype names in the figures (Fig 3, 4, 5, S1, S2 and S3), tables (S1, S2), throughout the text.

Comment #13

L289, Appendix Tables: When reporting R2 and F values we really don’t need 8 digits. I think 2 are enough. For p and p.adj values, the authors can also use a generalized p<0.05, p<0.01, p<0.001 or if they prefer to give an exact value please make sure the number of digits is homogene throughout the manuscript (main text and tables).

We have revised how we report the R2 and F values, which now show consistently two decimal digits. We kept to three decimal digits for p and p.adj values. These changes were applied throughout the manuscript and tables.

Comment #14

Sorry to emphasize this again but either discuss species OR genotype and not both at the same time. The discussion used species and genotypes as if there were synonyms and this is not correct. The authors could provide a genotype name to P. evermanni for an easy fix.

Alternatively, this would be the place to shift the discussion from genotype to species by assigning each genotype to a tentative species name (Porites evermanni, Porites cf/aff/X or Porites sp1, etc.), and use the species name throughout the manuscript.

L420: ‘The two cryptic P.lobata species (SSH2_plob and SSH3_plob)’ is a perfect example of what should be avoided.

‘The cryptic species P. lobata, which includes 2 genetic variants SSH2_plob and SSH3_plob’ would be more correct. There is probably more ways to turn this sentence.

Thank you for emphasizing this issue. We have provided answers and edits following comments #6, #11, and #12.

Comment #15

L424: I would also add that the findings for Porites should be further tested to include more specimens and morphologies.

This point was added to the Porites discussion paragraph together with edits related to comment #9.

(L472 to 481) “Porites corals were sampled based on the massive P. lobata morphotype. The presence of a taxonomy expert underwater at the time of sampling was not always possible. As a result, only three morphologically similar genetic lineages were included in the Porites data set. The addition of more species and morphotypes, from a wider range of geographical locations and environmental conditions should be considered to ascertain robustness of the derived findings. Nonetheless, the models showed promising results to resolve two cryptic P. lobata genetic lineages (SSH2_plob and SSH3_plob) within P. lobata species, which were also found in sympatry [5], since they were successfully classified to their corresponding genetic lineages by the RF models.”

L448: I haven’t seen any discussion on the potential bias that this study could have produced. For example, were specimens always collected from shallow, protected reefs? Or were they collected from various locations encompassing large environmental gradients? Also, the sampling design was targeting 1 particular morphology/morphotype/species, right? If this issue was probably solved for Pocillopora, I think the authors should add a few lines on this potential bias for Porites.

This point was added to the Porites discussion paragraph together with edits related to comment #9 and #15.

(L472 to 481) “Porites corals were sampled based on the massive P. lobata morphotype. The presence of a taxonomy expert underwater at the time of sampling was not always possible. As a result, only three morphologically similar genetic lineages

---

## [Decision Letter · Decision Letter 1]

Morphological traits and machine learning for genetic lineage prediction of two reef-building corals

PONE-D-25-02715R1

Dear Dr. Mitushasi,

We’re pleased to inform you that your manuscript has been judged scientifically suitable for publication and will be formally accepted for publication once it meets all outstanding technical requirements.

Kind regards,

Shashank Keshavmurthy, PhD

Academic Editor

PLOS ONE

Additional Editor Comments (optional):

Dear Authors

sorry for the delay in getting the review of your manuscript

I have now with me the latest review of your work and with some minor changes, it can be accepted and hence my decision to accept your work

best of luck

sincerely

shashank

Reviewers' comments:

Reviewer's Responses to Questions

**Comments to the Author**

Reviewer #1: All comments have been addressed

2. Is the manuscript technically sound, and do the data support the conclusions?

Reviewer #1: Yes

3. Has the statistical analysis been performed appropriately and rigorously?

Reviewer #1: Yes

4. Have the authors made all data underlying the findings in their manuscript fully available?

Reviewer #1: Yes

5. Is the manuscript presented in an intelligible fashion and written in standard English?

Reviewer #1: Yes

Reviewer #1: This is my second review of the manuscript written by Guinther Mitushisi and collaborators.

The authors did improve their manuscript and even added more analyses to support their findings. I still have a few comments which I think could improve their manuscript further. However, at this stage I believe most of these comments are minor and reflect a personal view rather than anything else.

Abstract

L55: I would re-order as follows: coral distribution < life history < ecology < evolution.

L59-60: Morphological traits were documented for Porites and Pocillopora corals species that were collected and genotyped […].

L62: I think some information is missing here to make the parallel between Porites and Pocillopora sampling. Suggestion: While Porites only included 3 tentative species, most Pocillopora spp. were accounted by included specimens from the western Indian Ocean, tropical Southwestern Pacific, and southeast Polynesia.

L71: correct species or correct genotype/lineage?

Introduction

L84: I would re-order as follows: physical and behavioral interactions < life history < adaptations < evolution.

L114-116: I think the authors could remove the “e.g.” and just leave the reference number here. I would do this throughout the manuscript.

L124: I would re-order as follows: phylogeny < coral evolution.

L142: Since two (out of three) lineages are cryptic I would still use lineages or genotypes instead of species here.

L145-153: In order to save some space and maybe gain clarity, I would start with genetic lineages [sensu reference] (corresponding to species names or tentative name [sensu reference]). The authors can just use the reference number (with no name). This will shorten the text and should be a little bit clearer. The authors can keep the original text when they want to give more context or propose a hybrid status to the lineage (L157).

L157-159: I think this is redundant and can be removed.

L160-161: Redundant with what follows. I suggest removing this sentence.

L163: I would suggest removing “non-invasive” here. As a reader, I sometimes got the impression that the authors wanted to test invasive vs noninvasive methods, or to support a field identification SOP using combined models (which corallite parameters are supporting the best corallum parameters) and detailed skeleton images… which is very interesting but not really what the authors are trying to achieve.

Material and methods

L205: It could also be “Porites morphometric measurements were done following Forsman et al. [23].”

L208-210: I’m sorry… in my first review I meant to combine figures and their respective table together. I think figures 1 and 2 are really pretty, and the use of tables is not doing them any justice. Do the authors think it could be possible to have the table info as legend of their respective figure?

Discussion

L459: I guess it’s growth form and not from, right?

L483: using RF models.

**Do you want your identity to be public for this peer review?** For information about this choice, including consent withdrawal, please see our Privacy Policy

Reviewer #1: No

---

## [Editor Report · Acceptance letter]

PONE-D-25-02715R1

PLOS ONE

Dear Dr. Mitushasi,

I'm pleased to inform you that your manuscript has been deemed suitable for publication in PLOS ONE. Congratulations! Your manuscript is now being handed over to our production team.

Kind regards,

on behalf of

Dr. Shashank Keshavmurthy

Academic Editor

PLOS ONE